The effect of aerobic and high-intensity interval training on plasma pentraxin 3 and lipid parameters in overweight and obese women

Cicek Guner 1
Ozcan Oguzhan 2
Akyol Pelin 3
Isik Ozkan 4
Novak Dario dario.novak@kif.unizg.hr 5
Küçük Hamza 6
1 Faculty of Sports Sciences, Hitit University , Corum , Turkey
2 Department of Biochemistry, Faculty of Medicine, Hatay Mustafa Kemal University , Hatay , Turkey
3 Faculty of Education, Department of Physical Education and Sports, Ondokuz Mayıs University , Samsun , Turkey
4 Faculty of Sports Sciences, Balıkesir University , Balıkesir , Turkey
5 Faculty of Kinesiology, University of Zagreb , Zagreb , Croatia
6 Yasar Doğu Faculty of Sport Sciences, Ondokuz Mayıs University , Samsun , Turkey
Jimenez Manuel
Electronic publication date: 2024 Sep 27
Publication date: 2024
Volume: 12
Electronic Location ID: e18123
Received 2024 Mar 15; Accepted 2024 Aug 28
Copyright: ©2024 Cicek et al.
Copyright year: 2024
Copyright holder: Cicek et al.
License: This is an open access article distributed under the terms of the Creative Commons Attribution License, which permits unrestricted use, distribution, reproduction and adaptation in any medium and for any purpose provided that it is properly attributed. For attribution, the original author(s), title, publication source (PeerJ) and either DOI or URL of the article must be cited.
License URL: https://creativecommons.org/licenses/by/4.0/

Keywords: Aerobic exercise, High-intensity interval training, Obesity, Pentraxin 3

Funding: The authors received no funding for this work.

==============================
Background

It is unclear whether different exercise programs lead to an increase in the concentration of plasma Pentraxin3 (PTX3), an anti-inflammatory protein. This study aimed to investigate the effects of aerobic exercise (AE) and high-intensity interval training (HIIT) on plasma PTX3 levels in overweight and obese women.

Method

A total of 45 sedentary women aged between 32.26 ± 6.30 voluntarily participated in the study. The control group (CG, n = 15) was selected among normal-weight women. Women in the group of participants who partook in exercise consisted of overweight and obese women according to a random method, including the AE group (n = 15) and the HIIT group (n = 15). The AE session conducted was 50 min in duration and consisted of warm-up exercises (5 min), and primary exercises (40 min, basic aerobic-step exercises). HIIT consists of warm-up exercises (5 min), primary exercises (work intervals: 6−10 × 1 min (80–90% HRmax), rest intervals: 1 min (walk, 50% HRmax), 21–29 min running. The exercises were applied for three sessions/week for 12 weeks. Fasting blood samples were taken from all participants before and after exercise and their body composition was measured.

Results

As a result of two different 12-week exercises, serum PTX3 levels increased significantly by 47.53% in the AE group and 50.21% in the HIIT group (p < 0.01). It was determined that the mean PTX3 before and after exercise increased from 1.71 ± 0.43 to 2.47 ± 0.40 ng/dL and HIIT from 1.62 ± 0.39 to 2.31 ± 0.33 ng/dL. A significant decrease in body mass index (BMI) values were detected, approximately 5.81% in the AE group and 5.06% in the HIIT group (p < .01). A significant decrease was detected in glucose, insulin, HOMA-IR, LDL-C, and hsCRP whereas HDL-C and VO2max value increased significantly in both exercise groups (p < .05; p < .01). There were no significant differences in TG and TC levels between groups (p > .05). Also, no significant differences were found between the two types of exercises in terms of parameters. A significant negative correlation in the total sample was found between PTX3 with BMI, fat mass, LDL-C, and hsCRP.

Conclusion

The percentage change in PTX3 values was not different between exercise types, whereas PTX3 was increased with exercise, regardless of the type of exercise. It can be said that both aerobic and HIIT increase PTX3, VO2max levels and improve lipid metabolism in overweight and obese women.

Introduction

Obesity is a chronic, progressive, complex, and heterogeneous disease with increased risk of multiple conditions that remains a global health problem, it is associated with increased morbidity, mortality, and reduced quality of life (Guerreiro, Carvalho & Freitas, 2022; Blüher et al., 2023). Obesity is characterized as a low-grade pro-inflammatory chronic condition by the enlargement of adipocytes and increased infiltration of circulating monocytes that differentiate into resident adipose tissue macrophages (Slusher & Huang, 2016). This pro-inflammatory state directly contributes to the increased risk and pathology of obesity-associated metabolic dysfunction, including the development of insulin resistance and subsequent type 2 diabetes mellitus (Patel, Buras & Balasubramanyam, 2013; Slusher, Huang & Acevedo, 2017). It is also well known that excessive fat accumulation in obese individuals is characterized by chronic and increased systemic inflammation, leading to various diseases, including cardiovascular diseases (Kasai et al., 2011; Zempo-Miyaki et al., 2019; Yu et al., 2021).

Insulin resistance, fat accumulation, and inflammation in these tissues characterize metabolic diseases, such as T2D and obesity (Lim & Kim, 2023). Some studies show the positive effect of different types of exercise and physical activity on the regulation of lipids, especially in diabetics (Saghebjoo et al., 2018; Akbarpour, Shoorabeh & Faraji, 2019). Exercise is known to improve the metabolic profile in various disorders by positively affecting insulin sensitivity and glucose metabolism, reducing plasma low-density lipoprotein cholesterol (LDL-C), triglyceride (TG), and total cholesterol (TC) concentrations while increasing high-density lipoprotein cholesterol (HDL-C) levels as well as reducing inflammation (Thyfault & Bergouignan, 2020; Soltani et al., 2023; Doewes et al., 2023).

Aerobic exercise (AE) training is an effective therapeutic approach used against obesity-related proinflammatory and metabolic dysfunction (Slusher & Huang, 2016). High-intensity interval training (HIIT) protocols vary and typically include a short, repetitive phase of high-intensity exercise immediately followed by a period of rest or low-intensity exercise (Valinejad, Omidi & Kalani, 2022). HIIT-type programs are used effectively and safely in different populations, including individuals at risk of chronic disease (Gibala et al., 2012). Many studies suggest that AE and HIIT programs are an economical and effective type of exercise because they can be easily applied to obese individuals and provide high energy expenditure, which is important in losing weight and burning fat (Racil et al., 2013; Petridou, Siopi & Mougios, 2019; Valinejad, Omidi & Kalani, 2022). It is also known that AE and HIIT can improve cardiovascular fitness in patients with coronary heart disease, people with metabolic syndrome, and people who are obese, as well as releasing anti-inflammatory markers and improving blood lipid panels, metabolic profiles, body composition, and quality of life (Jelleyman et al., 2015; Said et al., 2018; Gyorkos et al., 2019; Atakan et al., 2021).

PTX3 is an acute-phase protein that is similar to C-reactive protein (CRP) in both structure and function and belongs to the same family (Karamfilova et al., 2022). The pentraxin family includes two sets of long pentraxins and short pentraxins. Examples of short pentraxin include CRP produced in the liver and serum amyloid. A complete example of the long pentraxin family is PTX3 (Docherty et al., 2022; Valinejad, Omidi & Kalani, 2022). PTX3, an acute phase response protein, plays an anti-inflammatory role in obesity-related inflammation (Slusher, Huang & Acevedo, 2017). PTX3 plays a role in maintaining vascular tone and cardiovascular function and is also associated with vascular inflammation (Ristagno et al., 2019). Immune cells, such as monocytes and neutrophils, are major sources of PTX3 but also smooth muscle cells, mesangial cells, fibroblasts, and myeloid dendritic cells contribute to PTX3 production (Witasp et al., 2014).

Most studies report that plasma PTX3 concentrations are lower in obese individuals and those with metabolic dysregulation compared to normal weight and metabolically healthy controls (Chu et al., 2012; Miyaki et al., 2013; Slusher et al., 2021). However, studies on the effects of regular exercise on plasma PTX3 concentration and insulin resistance report conflicting results. High-intensity exercise increases PTX3 levels in healthy middle-aged men (Nakajima et al., 2010). Miyaki et al. (2013) found that there was an increase in PTX3 levels in middle-aged people after 2 months of aerobic exercise such as walking and cycling (Miyaki et al., 2013), another study found that plasma PTX3 concentration increased after 8 weeks of moderate-intensity aerobic endurance exercise in the elderly (Miyaki et al., 2012) while in other studies aerobic exercise has been shown to decrease PTX3 levels (Fukuda et al., 2012) and HIIT does not change PTX3 levels (Hovsepian et al., 2019). There is limited information regarding the potential effects of comparing aerobic and HIIT programs on PTX3 in obese individuals.

In this study, the primary hypothesis is that increased plasma PTX3 concentrations following 12 weeks of aerobic and HIIT would be positively associated with improved overweight and obese participants. The secondary hypothesis was that weight reduction and increased PTX3 were positively associated with lipid profile and VO2max in overweight and obese women after both exercise programs.

Materials & Methods

Study design and subject

This study was carried out in the province of Corum in Turkey as a randomized control trial with parallel group design according to CONSORT guidelines. The study was conducted between April and September 2022. The Clinical Research Ethics Committee of the Medical Faculty of Ondokuz Mayis University approved the study (2020/87) in accordance with the policy statement of the Turkey Ministry of Health. This study was approved by the Local Ethical Committee and performed according to the Declaration of Helsinki. All participant completed a medical questionnaire and were informed about the possible risks and discomfort involved before giving their written consent to participant.

Participants were recruited via posters, word of mouth, social media, and email around the academic community of Hitit University in Türkiye. A total of 100 participants responded to the advertisement and were invited to the campus for a face-to-face interview. In this study, while the sedentary women (MeanAge = 34.20 ± 4.16 years) in the control group had a normal body mass index, the sedentary women in the experimental group consisted of obese and overweight women. Sedentary women in the experimental group were randomly divided into two groups: AE groups (MeanAge = 32.93  ± 6.61 years) and HIIT groups (MeanAge = 29.67 ± 7.22 years).

The eligibility criteria were women between the ages of 20 and 40 years and sedentary (to practice <20 minutes of moderate-vigorous physical activity on <3 days/week, not to have been involved in an exercise training program in the previous 3 months) and the individuals involved were invited to participate in the study. Women were included if they had a body mass index (BMI) between 18.5 and 35 kg/m2 (normal, overweight to obese BMI categories). All participants had health examinations to ensure that they were not taking any medication for lipid metabolism or anti-pregnancy drugs or smoking cigarettes and were not suffering from any kind of heart disease, diabetes, hypertension, respiratory, metabolic, or inflammatory disorders. Individuals with any of the following characteristics were excluded from the study: going through menopause, having a smoking metabolic or cardiovascular disease, arrhythmias, heart failure, hypertension, diabetes mellitus, orthopedic limitations, exercising regularly for the last 6 months, and having a BMI below 18.5 and over 34.9 kg/m2. After the interview 55 participants were not eligible for the study due to excluded criteria and 45 potential participants were asked to sign the written consent provided to them.

Table 1 shows that the pre-test average body mass indexes of sedentary women in the aerobic exercise, HIIT, and control groups were 31.65 ± 4.64 kg/m2, 29.77 ± 4.64 kg/m2, and 21.38 ± 1.17 kg/m2, respectively. Considering the pre-BMI values of the participants, it can be understood that the control group consisted of normal-weight women and the exercise groups consisted of overweight and obese women. Moreover, there was no statistically significant difference between the BMI values of the exercise groups (AE and HIIT).

Table 1 Body mass index results of sedentary women.

Variables	Groups	n	Pre-test	F	p	
BMI (kg/m2)	AE	15	31.65 ± 4.64a	30.236	0.01**	
HIIT	15	29.77 ± 4.64a	
Control	15	21.38 ± 1.17b	
Notes.

** p < 01

abc one-way ANOVA

a,b,c Different letter shows statistical differences between groups with the LSD test.

Randomization of subjects

In the study, a total of 45 women were divided into groups. The CG (n = 15) was selected as normal weight (BMI; 18.5–24.9 kg/m2) participants and randomization was not included. The remaining 38 participants (BMI; 25–34.9 kg/m2) were listed as overweight and obese and would be undertaking exercise, these participants were randomly selected. The thirty-eight subjects in the overweight and obesity category started the run-in period and following this were randomized into two groups: They were randomly allocated to either the AE (n = 19) or HIIT (n = 19) group for the 12-week program (permuted-block randomization conducted using a computer-generated random allocation). The sequences were generated by an investigator who was not involved in the data collection procedures. Upon enrollment, the group allocation of the participant was revealed to a non-blinded research assistant responsible for data collection.

Blinding

The researcher collecting data was not blinded to the group in which each participant is allocated because they took all pre-during, and post-intervention measures from the participants and monitored adherence to the AE and HIIT intervention for the duration of the 12 weeks. The assignment of a greater number of participants to the intervention groups was due to the risk of dropouts. The flow diagram showing the patients participating in the study is shown in Fig. 1.

Figure 1 Flowchart of the participants.

Calculation of the sample size

A power analysis was performed to determine the sample size for generalizing the research results. The total number required to find the expectation of a medium effect size (f = 0.60) to be statistically significant in revealing the effect of the three different groups was determined as 45 (fifteen participants for each group; α = 0.05; 1 − β = 0.80). It is recommended that the effect size value be ≥.5 in clinical studies (Kilic, 2014).

Experimental design

The research was designed with a true experimental design and the participants were divided into three different groups as the CG, AE, and HIIT exercise groups. At the beginning of the study, the participants’ anthropometric, body composition measurements, and lipid profiles were measured. After the pre-test measurements, the participants in the experimental groups applied a total of 36 sessions of exercise 3 days a week (Monday, Wednesday, and Friday) for 12 weeks. Apart from the training, all participants were instructed to maintain the physical activity that they had been practicing without any caloric restriction. During the exercise, in this group, participants were asked to continue their present lifestyle habits and not change anything in their daily routine (e.g., exercise, nutrition). At the baseline, and after the 12 weeks of physical training, anthropometric, body composition measurements, and lipid profile parameters were performed in all of the groups.

Exercise training protocols

A total of 36 training sessions were carried out for each group. All exercise sessions were supervised by the same trained physical education professional. Training was performed under the supervision of a qualified and certified fitness instructor. Aerobic exercise intensity was increased progressively from HR 60% to 80% (between the 1st and 4th week was HR 60%, between the 5th and 8th week was HR 70% and 9th and 12th was HR 80%). Each exercise session lasted 50 min in duration and consisted of warm-up exercises (5 min), and primary exercises (40 min, basic aerobic-step exercises). HIIT training adapted from (Poon et al., 2020) consisted of warm-up exercises (5 min), primary exercises (work intervals: 6–10 × 1 min (80–90% HRmax), rest intervals: 1 min (walk, 50% HRmax), 21–29 min running on the treadmill where the target heart rate (THR) was controlled using a heart rate monitor for each subject (A 360; Polar Electro, Kempele, Finland) during each exercise session to monitor exercise intensity. The heart rate monitors stored all exercise session data (including HR and estimated energy expenditure) and were used to confirm participants’ compliance with the exercise prescription three times per week at the end of the 12-week intervention.

Procedure and Measurement

Anthropometrics and body composition

Participants were instructed to stand upright, wearing light clothing and barefoot. Height was measured with an accuracy of 0.1 cm using a portable stadiometer (model Seca 215 brand) and weight was measured to the nearest 0.1 kg using an electronic scale. BMI was calculated as weight in kilograms divided by height squared (kg/m2) in meters (Mackenzie, 2005). The participants were asked to breathe out for measurement of their waist circumference, which was measured to the nearest 0.1 cm at the iliac crest (Tape measure). When viewed from the sagittal plane, hip circumference was evaluated at the level of the maximum extension of the thigh, and the waist-hip ratio equals the waist circumference divided by the hip circumference (Pua & Ong, 2005). The fat mass of the participants were measured in sedentary mode with TANITA BC-418 (Tanita Corp., Tokyo, Japan) brand bioelectrical impedance analyzer in the morning on an empty stomach, wearing shorts and a t-shirt.

Maximal oxygen uptake (VO2max)

VO2max was determined by a 20 m shuttle run test. For this test, the Powertimer PC 1.9.5 Version Newest device was used. This consisted of a shuttle run between two met alters placed 20 m apart at increasing speeds. Two photocells were placed on the starting and ending at the 20-meter distance. The VO2max value was determined in ml/kg/min based on the number of shuttle runs executed.

Blood biochemistry

Morning venous blood from the forearm was collected in 10 cc biochemistry tubes after at least 8 h of fasting during the pre- and post-exercise periods. All samples were centrifuged at 1,500× g for 10 min and their serum was separated. Serum glucose, TC, TG, HDL-C, and LDL-C levels were measured by the spectrophotometric method in the Biochemistry Device (Siemens Advia 1800), serum insulin level was measured by the chemoluminescent method in the hormone device (Siemens Advia Centaur XP) and serum hsCRP measurements were made by the nephelometric method in the Nephelometer Device (Siemens BNII). Insulin resistance (IR) was determined by HOMA-IR, calculated from plasma insulin and glucose values according to HOMA-IR = Fasting insulin (µU/ml) x fasting glucose (mg/dL)/405. A Thermo Scientific/MultiscanGo UV (USA) device was used for the parameters measured by the ELISA method, and a washer (Thermo Scientific, Vantaa, Finland) device was used for the washing steps in the protocols. Working with the ELISA method for serum Pentraxin quantification (FINETEST Human Pentraxin 3 ELISA Kit, Cat. No: EH0263) a brand commercial kit was used. Optical density was measured spectrophotometrically at a wavelength of 450 nm. Concentrations were calculated using the 4PL (four-parameter logistic curve) calibration curve. Results are expressed as ng/mL. The analysis range was determined as 0.313-20 ng/mL and the sensitivity was 0.188 ng/mL. Precision, intra-assay CV (coefficient of variation); 8%, inter-assay CV; It is 10%. All tests were performed at Hatay Mustafa Kemal University Hospital Central and Research Laboratory.

Statistical analysis

IBM SPSS Statistics 24 software was used for the statistical analysis. The normality of the data obtained was tested with the Shapiro–Wilk test and it was observed that it was normally distributed. Mean and standard deviation values were used as descriptive statistics. Percentage change analysis was preferred because the pre-test values of the study groups classified as randomized controlled were not homogeneous. The percentage changes of the measured variables were calculated with the formula “% Δ = (Pretest − Posttest)/Pretest*100”. Pearson correlation analysis was used to determine the relationship between the percentage difference between the measurement variables. Additionally, one-way ANOVA with the delta approach was used to analyze the obtained data. The LSD post-hoc test was preferred to determine the source of the percentage difference between the groups. The confidence interval was chosen as 95% and values below p < 0.05 were considered statistically significant.

Results

The averages of descriptive statistics of pre and post-test measurements of physical and physiological variables in different exercise groups were given in Table 2.

Table 2 Descriptive statistics of pre- and post-test measurements of physical and physiological variables in different exercise groups.

Variables	Groups	n	Pre-Test	Post-Test	
Body Weight (kg)	AE	15	80.23 ± 10.58	75.58 ± 10.00	
HIIT	15	77.29 ± 10.33	73.47 ± 10.62	
Control	15	54.52 ± 2.31	54.29 ± 2.42	
BMI (kg/m2)	AE	15	31.65 ± 4.64	29.83 ± 4.60	
HIIT	15	29.77 ± 4.64	28.29 ± 4.65	
Control	15	21.38 ± 1.17	21.29 ± 1.17	
Fat Mass (kg)	AE	15	39.04 ± 3.46	35.87 ± 3.44	
HIIT	15	38.61 ± 4.05	35.16 ± 4.16	
Control	15	23.21 ± 3.49	22.73 ± 3.45	
Waist Circumference (cm)	AE	15	95.40 ± 10.97	93.07 ± 10.66	
HIIT	15	96.00 ± 9.83	92.13 ± 8.98	
Control	15	68.80 ± 3.34	68.73 ± 3.34	
Hip Circumference (cm)	AE	15	112.60 ± 6.17	109.20 ± 6.04	
HIIT	15	115.60 ± 7.41	111.40 ± 7.98	
Control	15	96.33 ± 3.44	96.33 ± 3.89	
Waits-Hip Ratio (cm)	AE	15	0.85 ± 0.07	0.85 ± 0.07	
HIIT	15	0.83 ± 0.06	0.83 ± 0.06	
Control	15	0.71 ± 0.04	0.71 ± 0.04	
VO2max (mL/kg/min)	AE	15	24.91 ± 1.07	31.21 ± 2.10	
HIIT	15	24.93 ± 0.94	30.72 ± 2.21	
Control	15	25.53 ± 1.85	26.29 ± 1.49	
Notes.

BMI body mass index

WHR waist/hip ratio

VO2max maximal oxygen uptake

AE aerobic exercise

HIIT high intensity interval training

When Table 3 was examined, there was no statistical effect of 12 weeks of regular AE and HIIT exercises on the waist-hip ratio. On the other hand, it was found that statistically significant differences on body weight, BMI, fat mass, waist circumference, hip circumference, and VO2max. According to this result, following 12 weeks of AE and HIIT exercises, a significant percentage decrease was seen in the BMI, fat mass, hip circumference, and an increase in VO2max in overweight and obese women compared to the control group, and there was no statistical difference between the exercise groups (AE and HIIT). Moreover, when the differences in waist circumference were examined, the percentage change in all groups was different from each other and HIIT training was found to be the most effective exercise model in reducing waist circumference.

Table 3 Comparison of percentage changes in physical and physiological variables in different exercise groups.

Variables	Groups	n	ΔMean ± S.D.	F	p	
Body Weight (kg)	AE	15	−5.81 ± 2.91a	23.756	0.001**	
HIIT	15	−5.06 ± 2.17a	
Control	15	−0.42 ± 1.71b	
BMI (kg/m2)	AE	15	−5.81 ± 2.91a	23.756	0.001**	
HIIT	15	−5.06 ± 2.17a	
Control	15	−0.42 ± 1.71b	
Fat Mass (kg)	AE	15	−8.09 ± 3.85a	13.861	0.001**	
HIIT	15	−9.01 ± 3.51a	
Control	15	−1.89 ± 4.64b	
Waist Circumference (cm)	AE	15	−2.44 ± 1.20b	19.029	0.001**	
HIIT	15	−3.97 ± 2.31a	
Control	15	−0.08 ± 1.51c	
Hip Circumference (cm)	AE	15	−3.02 ± 0.84a	20.638	0.01*	
HIIT	15	−3.65 ± 2.54a	
Control	15	−0.01 ± 1.04b	
Waits-Hip Ratio (cm)	AE	15	−0.60 ± 0.82	0.674	0.515	
HIIT	15	−0.27 ± 3.28	
Control	15	−0.07 ± 1.59	
VO2max (mL/kg/min)	AE	15	25.35 ± 7.68a	40.763	0.001**	
HIIT	15	23.37 ± 9.26a	
Control	15	3.17 ± 4.63b	
Notes.

* p < .05.

** p < 01

abc one-way ANOVA

a,b,c Different letter shows statistical differences between groups with the LSD test.

Values in bold indicate statistically significant difference.

The averages of descriptive statistics of pre and post-test measurements of glucose and lipid metabolism variables in different exercise groups were given in Table 4.

Table 4 Descriptive statistics of pre- and post-test measurements of glucose and lipid metabolism variables in different exercise groups.

Variables	Groups	n	Pre-Test	Post-Test	
PTX3 (ng/dL)	AE	15	1.71 ± 0.43	2.47 ± 0.40	
HIIT	15	1.62 ± 0.39	2.31 ± 0.33	
Control	15	2.98 ± 0.44	2.98 ± 0.41	
HOMA-IR (Unit)	AE	15	2.74 ± 0.73	2.15 ± 0.74	
HIIT	15	2.69 ± 0.75	2.01 ± 0.70	
Control	15	1.42 ± 0.45	1.39 ± 0.41	
HsCRP (mg/dL)	AE	15	2.46 ± 1.02	1.40 ± 0.52	
HIIT	15	2.64 ± 1.03	1.37 ± 0.69	
Control	15	1.09 ± 0.56	1.11 ± 0.66	
Glucose (mg/dL)	AE	15	95.13 ± 8.61	88.80 ± 7.60	
HIIT	15	97,40 ± 8.30	89.80 ± 6.28	
Control	15	78,67 ± 5.00	78.93 ± 4.93	
Insulin (U/mL)	AE	15	11.82 ± 2.84	8.90 ± 2.01	
HIIT	15	12.09 ± 2.97	9.52 ± 4.30	
Control	15	7.23 ± 1.96	7.24 ± 1.96	
LDL-C (mg/dL)	AE	15	168.73 ± 30.18	151.40 ± 18.49	
HIIT	15	161.47 ± 16.43	144.40 ± 17.05	
Control	15	119.67 ± 14.63	120.47 ± 14.21	
HDL-C (mg/dL)	AE	15	41.93 ± 11.88	50.13 ± 10.57	
HIIT	15	44.23 ± 8.56	49.26 ± 8.43	
Control	15	52.07 ± 7.04	51.78 ± 6.75	
TC (mg/dL)	AE	15	173.13 ± 29.49	155.20 ± 23.63	
HIIT	15	170.87 ± 39.95	148.87 ± 20.73	
Control	15	136.40 ± 13.39	137.47 ± 13.73	
TG (mg/dL)	AE	15	136.47 ± 6897	122.87 ± 50.18	
HIIT	15	124.67 ± 53.63	109.20 ± 32.19	
Control	15	72.60 ± 21.19	73.87 ± 19.57	
Notes.

HOMO-IR Homeostatic Model Assessment for Insulin Resistance

TC Total cholesterol

TG Triglyceride

HDL-C high density lipoprotein-cholesterol

LDL-C Low density lipoprotein-cholesterol

PTX3 Pentraxin 3

Hs-CRP high-sensitivity C-reactive protein

When Table 5 was examined, there was no statistical effect of 12 weeks of regular aerobic and HIIT exercises on TC and TG levels. On the other hand, it was found that statistically significant differences in PTX3, HOMA-IR, hsCRP, glucose, insulin, LDL-C, and HDL-C levels. According to this result, following 12 weeks of aerobic and HIIT exercises, a significant percentage decrease was seen in the HOMA-IR, hsCRP, glucose, insulin, LDL-C and an increase in PTX3 and HDL-C of overweight and obese women compared to the control group, and there was no statistical difference between the exercise groups (AE and HIIT).

Table 5 Comparison of percentage changes in glucose and lipid metabolism variables in different exercise groups.

Variables	Groups	n	ΔMean ±S.D.	F	p	
PTX3 (ng/dL)	AE	15	47.53 ±25.69a	14.691	0.001**	
HIIT	15	50.21 ±41.81a	
Control	15	0.17 ±4.34b	
HOMA-IR (Unit)	AE	15	−17.35 ±29.25a	5.126	0.01*	
HIIT	15	−24.46 ±19.03a	
Control	15	−1.12 ±6.19b	
HsCRP (mg/dL)	AE	15	−37.16 ±26.85a	7.138	0.002**	
HIIT	15	−40.69 ±41.63a	
Control	15	7.74 ±46.28b	
Glucose (mg/dL)	AE	15	−6.05 ±10.64a	4.677	0.015*	
HIIT	15	−7.43 ±7.03a	
Control	15	0.36 ±1.96b	
Insulin (U/mL)	AE	15	−22.64 ±16.13a	4.122	0.023*	
HIIT	15	−19.06 ±37.52a	
Control	15	0.34 ±0.85b	
LDL-C (mg/dL)	AE	15	−8.05 ±18.00a	3.253	0.049*	
HIIT	15	−10.03 ±11.26a	
Control	15	0.74 ±1.99b	
HDL-C (mg/dL)	AE	15	24.79 ±28.79a	5.571	0.007**	
HIIT	15	13.55 ±21.29a	
Control	15	−0.46 ±3.16b	
TC (mg/dL)	AE	15	−8.82 ±15.78	2.611	0.085	
HIIT	15	−9.80 ±18.33	
Control	15	0.78 ±1.92	
TG (mg/dL)	AE	15	5.98 ±55.55	0.329	7.22	
HIIT	15	−4.62 ±29.77	
Control	15	2.53 ±4.97	
Notes.

* p < 0.05.

** p < 01.

abc one-way ANOVA

a,b,c Different letter shows statistical differences between groups with the LSD test.

Values in bold indicate statistically significant difference.

When Table 6 was examined, the relationship between the variables were as follows.

Table 6 Relationships between measured variables.

Variables		ΔBody Weight	Δ BMI	ΔFat Mass	ΔVO 2max	ΔPTX3	Δ HOMA-IR	ΔHsCRP	ΔGlucose	ΔInsulin	Δ LDL-C	ΔHDL-C	ΔTC	
Δ Body Weight (kg)	r	1												
	p													
Δ BMI (kg/m 2 )	r	0.829**	1											
	p	0.001												
Δ Fat Mass (kg)	r	0.441**	0.531**	1										
	p	0.002	0.001											
Δ VO 2max (mL/kg/min)	r	−0.433**	−0.538**	−0.471**	1									
	p	0.003	0.001	0.001										
Δ PTX3 (ng/dL)	r	−0.362*	−0.507**	−0.400**	0.463**	1								
	p	0.015	0.001	0.006	0.001									
Δ HOMA-IR (Unit)	r	0.360*	0.348*	0.305*	−0.309*	−0.221	1							
	p	0.015	0.019	0.042	0.039	0.145								
Δ HsCRP (mg/dL)	r	0.308*	0.351*	0.258	−0.491**	−0.333*	0.151	1						
	p	0.039	0.018	0.088	0.001	0.026	0.322							
Δ Glucose (mg/dL)	r	0.353*	0.606**	0.245	−0.466**	−0.294	0.143	0.232	1					
	p	0.017	0.001	0.105	0.001	0.050	0.348	0.125						
Δ Insulin (U/mL)	r	0.291*	0.295*	0.335*	−0.475**	−0.189	0.505**	0.136	0.152	1				
	p	0.0.50	0.049	0.024	0.001	0.214	0.001	0.373	0.320					
Δ LDL-C (mg/dL)	r	0.364*	0.401**	0.139	−0.464**	−0.388**	0.042	0.186	0.675**	0.043	1			
	p	0.013	0.006	0.361	0.001	0.009	0.786	0.222	0.001	0.778				
Δ HDL-C (mg/dL)	r	0.016	−0.135	−0.255	0.249	0.330*	−0.098	−0.158	0.011	−0.226	0.190	1		
	p	0.915	0.378	0.091	0.099	0.027	0.523	0.299	0.941	0.136	0.212			
Δ TC (mg/dL)	r	0.297*	0.319*	0.316*	−0.187	−0.204	0.157	0.021	0.082	0.084	−0.023	0.007	1	
	p	0.047	0.033	0.034	0.219	0.179	0.302	0.891	0.592	0.584	0.882	0.965		
Δ TG (mg/dL)	r	0.004	0.030	0.164	−0.030	0.192	−0.026	0.134	0.009	−0.004	−0.014	−0.020	0.340*	
	p	0.680	0.843	0.282	0.842	0.206	0.863	0.382	0.955	0.978	0.927	0.894	0.022	
Notes.

* p < 0.05

** p < 0.01.

N:45.

• There was a negative relationship between PTX3 with the percentage changes in hsCRP (r =  − .333) and LDL-C (r =  − .388) values, and a positive relationship between the percentage changes in HDL-C (r = .330) values.

• There was a negative relationship between body weight with the percentage changes in VO2max (r =  − .433) and PTX3 (r =  − .362) values, and a positive relationship between the percentage changes in BMI (r = .829), fat mass (r = .441), HOMA-IR (r = .360), hsCRP (r = .308), glucose (r = .353), insulin (r = .291), LDL-C (r = .364) and TC (r = .297) values.

• There was a negative relationship between BMI with the percentage changes in VO2max (r =  − .538) and PTX3 (r =  − .507) values, and a positive relationship between the percentage changes in fat mass (r = .531), HOMA-IR (r = .348), hsCRP (r = .351), glucose (r = .606), insulin (r = .295), LDL-C (r = .401) and TC (r = .319) values.

• There was a positive relationship between fat mass with the percentage changes in HOMA-IR (r = .305), insulin (r = .335), and TC (r = .316) values, and a negative relationship between the percentage changes in VO2max (r =  − .471) and PTX3 (r =  − .400) values.

• There was a positive relationship between the percentage changes in VO2max with PTX3 (r = .463) values and a negative relationship between the percentage changes in HOMA-IR (r =  − .309), hsCRP (r =  − .491), glucose (r =  − .466), insulin (r =  − .475) and LDL-C (r =  − .464) values.

• There was a positive relationship between HOMA-IR and percentage changes in insulin (r = .505) values.

• There was a positive relationship between the percentage changes in glucose and LDL-C (r = .675) values.

• There was a positive relationship between the percentage changes in TC and TG (r = .340) values.

Discussion

This study aimed to increase the plasma PTX3 value in overweight and obese women following a 12-week AE and HIIT program. As a result of both exercise programs, it was determined that weight loss had a positive effect on PTX3 and lipid profiles in women in the AE and HIIT groups. Additionally, a positive correlation was found between PTX3 with HDL-C, a negative correlation between PTX3 with LDL-C and hsCRP and it was also found to improve cardiometabolic health.

In the present study, pre-exercise PTX3 values of the subjects in the AE and HIIT groups were found to be lower than those in the normal weight control group (Table 4). However, after 12 weeks of exercise, it was determined that there was a statistically significant increase in PTX 3 by 48% in the AE group and 50% in the HIIT group (Table 5). Various reasons have been proposed for the exercise-induced increase in PTX3. Several researchers have emphasized the significance of weight loss in increasing circulating PTX3 levels in previous studies (Witasp et al., 2014; Weiss et al., 2017). In this study, through comparing subjects that engaged in 12 weeks of AE and HIIT training with healthy control groups, a significant decrease was observed in BMI, body weight, fat mass, waist, and hip circumference values (p > 0.01). It was found that BMI percentage changes in exercise groups, a decrease in 6% AE and 5% in HIIT were observed (Table 3). Both exercise protocols showed approximately similar changes in both PTX3 and body composition parameters. A few studies explain whether weight loss with the effect of exercise will increase circulating PTX3 levels in overweight and obese individuals. In previous studies, aerobic and HIIT exercises applied to overweight and obese individuals have been found to significantly reduce BMI, body weight, fat mass, and waist and hip circumference (Park et al., 2020; Gallo-Villegas et al., 2022; Armstrong et al., 2022). Studies predominantly indicate that circulating levels of PTX3 in obese and overweight individuals are lower compared to those in normal-weight individuals. Circulating PTX3 concentrations are negatively correlated with various obesity indices, including BMI, waist and hip circumference, and visceral fat mass (Zempo-Miyaki et al., 2019; Osorio-Conles et al., 2011; Chu et al., 2012; Miyaki et al., 2012; Slusher, Huang & Acevedo, 2017). Regular exercise reduces body weight and fat mass in overweight and obese and may trigger metabolic adaptations that may lead to improved lipid profiles (Zouhal et al., 2020; Nazari, Minasian & Hovsepian, 2020). As a matter of fact, the current study’s negative correlation between PTX3 with BMI and fat mass supports this view (r =−0.507; p = 0.01, r =−0.400; p = 0.06). The primary hypothesis was confirmed in that regular exercise and weight loss positively increase PTX3 levels in overweight and obese women. In addition, PTX3 levels approached the level of healthy normal control group subjects. The effects of weight loss on PTX3 may vary from person to person and depend on several factors. For example, the speed of the weight loss process, type of diet, exercise programs (frequency, intensity, time, and type), etc.

Cardiorespiratory fitness has been shown to be a strong indicator of both cardiovascular and metabolic health (Kodama et al., 2009). VO2max is considered an indicator of cardiovascular health and cardiopulmonary fitness (Doijad, Kample & Surdi, 2013). Some studies have shown that improvements in cardiorespiratory fitness following an exercise intervention were associated with positive changes in other cardiovascular diseases and metabolic biomarkers in obese populations (Campbell et al., 2009; Slusher & Huang, 2016). Previous studies have confirmed that aerobic and HIIT training leads to an increase in VO2max levels in overweight and obese women, validating the positive impact of exercise on cardiovascular fitness, as observed in the increase in VO2max (Arad et al., 2015; Türk et al., 2017; Van Baak et al., 2021; Valsdottir et al., 2020; Mogharnasi, TaheriChadorneshin & Abbasi-Deloei, 2019). In contrast, a recent study (Arboleda-Serna et al., 2022) demonstrated that low HIIT did not have a superior effect on VO2max and cardiorespiratory fitness compared to moderate-intensity continuous training in overweight women. In this study, when comparing exercise groups with the control group, a significant increase of 25% and 23% in VO2max values were found in the AE and HIIT groups after 12 weeks of exercise, supporting previous studies (Table 3). Moreover, our results found a positive correlation between PTX3 and VO2max (r = 463, p = 0.01). A positive increase in PTX3 was detected with the increase in VO2max level. Moreover, these results confirmed that intervention methods that elevate plasma PTX3, such as AE and HIIT interventions aimed at enhancing VO2max, may provide therapeutic benefits and augment the cardioprotective and potential metabolic effects of PTX3 (Slusher & Huang, 2016).

Lipid profile disorders observed in obese individuals are closely linked to many inflammatory markers underlying atherosclerosis and are major risk factors for cardiovascular diseases (Nazari, Minasian & Hovsepian, 2020; Ortega, Lavie & Blair, 2016). This study found that both exercise groups exhibited a significant increase in HDL-C and a significant decrease in LDL-C compared to the control group after 12 weeks of exercise (p < 0.05 and p < 0.01). A significant positive correlation was detected between PTX3 and HDL-C (r = 0.330; p = 0.27) and a significant negative correlation between PTX3 and LDL-C (r =-388; p = 0.09). However, no difference was detected in terms of TC and TG levels and parameters between exercise groups (p > 0.01). In addition, the approximately 8–10% reduction in LDL-C in both AE and HIIT groups indicates a significant health benefit.

Moraleda et al. (2013) found that 24 weeks of strength, endurance, and combined exercise applied to obese individuals reduced LDL-C levels by 11.2%, 10.8%, and 7.9%, respectively. They also emphasized the importance of exercise in the treatment of obesity and related comorbidities. Moreover, we found that approximately 8–10% reduction in LDL-C in both AE and HIIT groups indicates a significant health benefit for overweight and obese women. This is noteworthy because even a 1% reduction in LDL is believed to decrease cardiovascular disease risk by 2%, and treatment goals for patients with atherosclerosis are determined based on LDL levels (Hovsepian et al., 2019). In fact, the significant decrease in LDL-C in both training groups was associated with an improvement in overall fat metabolism and significant improvements in insulin resistance. The control group did not undergo any significant alteration of any of the cited variables. On the other hand, regular exercise has been demonstrated in numerous studies to improve lipid profiles, resulting in reduced TG, LDL-C, and TC, along with increased HDL-C (Branco et al., 2019; Nazari, Minasian & Hovsepian, 2020). It has been shown that PTX3, especially released from endothelial cells, is stimulated by HDL-C and has been suggested as another reason for the increase in PTX3 (Norata et al., 2008).

AE and HIIT groups saw a significant decrease in glucose and insulin levels after the 12-week exercise program (p < 0.5). Moreover, this study showed that a significant reduction of 17% in HOMA-IR values in the AE group and 24% in the HIIT group, and subsequently approaching the values of the healthy control group and indicating improvement (Table 5). Exercise training programs were effective in reducing insulin resistance in patients with or without type 2 diabetes, overweight or obesity (Battista et al., 2021). Physical exercise may have regulated glucose metabolism not only in muscle but also in fat and liver tissue, resulting in an overall improvement in insulin sensitivity (Smith et al., 2018). Researchers noted that HIIT and aerobic exercise positively affect insulin resistance in overweight/obese populations (Batacan et al., 2017; Gallo-Villegas et al., 2022). Racil et al. (2013) concluded that improvements in body composition and lipid profiles were associated with cardiovascular adaptations, reducing HOMA-IR in young obese women. On the other hand, studies still present conflicting data regarding the association of PTX3 with adiposity and insulin resistance in various diseases (Hollan et al., 2010; Chu et al., 2012). The reason for this may be related to the number of subjects and the type of exercise applied. However, the current study found a weak negative association of PTX3 with insulin and HOMA-IR (Table 6). However, no correlation was detected between PTX3 and HOMA-IR. This study showed that the positive effects of PTX3 on the lipid profile are independent of insulin resistance. Therefore, the positive effect of a decrease in body weight on lipid profile and insulin resistance through regular exercise supports the positive increase in PTX3 level in this study. In this study, it was also found that it reduced inflammation by increasing PTX3 and improving lipid parameters in overweight and obese women, and also approached the levels of healthy normal weight control group subjects and improved overall metabolic health.

Another protein belonging to the Pentraxin family is CRP, which belongs to the short PTX subgroup and is known as an acute phase reactant released from the liver (Nakajima et al., 2010). In obesity, inflammation occurs through adipose tissue. Cytokines such as TNF- α and IL-6 released from adipose tissue stimulate the liver, leading to the release of CRP (Koçak & Öney, 2021). Obesity-associated inflammation is marked by increased serum CRP levels (Imayama et al., 2012; Bernhardt et al., 2022). In the current study, a significant decrease of 37% in hsCRP values in the AE group and 41% in the HIIT group was observed, supporting studies demonstrating exercise’s positive effects on hsCRP. Moreover, it was found that the hsCRP levels of the AE and HIIT groups approached the level of the control group subjects after 12 weeks of exercise. In contrast, Libardi et al. (2012) have reported that sixteen weeks of resistance, endurance, or concurrent exercises in middle-aged healthy men did not affect CRP levels. Additionally, Huffman et al. (2006) reported that exercise does not appear to significantly alter hsCRP concentrations in different-intensity aerobic exercise in sedentary overweight to mildly obese men and women. The different results in these studies may be due to the subjects and exercise types. On the other hand, some researchers have found that AE and HIIT reduce hsCRP levels in obese and overweight individuals (García-Hermoso et al., 2016; Cicek et al., 2017; Mohammadkhani et al., 2021). Another study found that 5% weight loss after a 12-month moderate-intensity aerobic exercise and diet program reduced hsCRP in overweight and obese postmenopausal women (Imayama et al., 2012). In parallel with previous literature, our study found that hsCRP levels decreased as a result of approximately 6% and 5% weight loss in the AE and HIIT groups. In addition, a significant weak positive correlation was found between BMI and fat mass with hsCRP (r = 0.351; p = 0.018, r = 258; p = 0.88). Along with the results from this study, this data indicates that aerobic and HIIT exercises lead to a decrease in hsCRP levels through significant reductions in BMI, representing an obesity index reflecting fat mass. Some researchers supporting our findings emphasize the necessity of reducing body fat mass, especially abdominal obesity, among the factors in reducing hsCRP after exercise (Jorge et al., 2011; Pestana et al., 2016).

Weight loss can reduce low-grade inflammation in the body. Inflammation can affect the levels of proteins produced by the immune system, such as PTX3. The present study found that exercise decreased hsCRP, one of the inflammation markers in obesity, and caused an increase in the level of PTX 3, which has anti-inflammatory properties.

Conclusions

In this study, PTX3 increased in overweight and obese women after 12 weeks of AE and HIIT. Furthermore, negative correlations were identified between PTX3 and BMI, fat mass, hsCRP, and LDL-C, while positive correlations were observed between PTX3 and VO2max and HDL-C. Additionally, a decrease in hsCRP was noted, while serum PTX3 levels significantly increased compared to the control group. However, no significant differences were found between the two types of exercises in terms of parameters. it was observed that both types of exercise led to significant improvements in body composition and notable enhancements in blood lipids and insulin resistance compared to the control group. This study suggests that weight loss following lifestyle changes significantly increased PTX3 levels in overweight and obese women. The implementation of aerobic and HIIT exercise protocols in overweight/obese populations is known to induce significant positive physiological adaptations, reducing the development and progression of obesity-related risk factors. Therefore, regular exercise participation is believed to improve cardiovascular and metabolic health by increasing resting concentrations of plasma PTX3.

Limitation

Participants were advised not to alter their daily physical activity and eating habits. However, their diets were not rigorously controlled, and the positive impact of exercise on behavioral eating may have led to a reduced calorie intake. Another limitation is that only one gender and one type of obesity were included in the study, with the control group consisting of women of average weight. In future studies, combined investigations of aerobic and HIIT exercises at different intensities and methods, such as calorie restriction, should be considered. Additionally, examining PTX3 levels based on obesity types for both genders is recommended for a more comprehensive and generalizable understanding of the results.

Supplemental Information

Supplemental Information 1 Data set excel

The authors thank to sedentary women for their participation in the study.

Additional Information and Declarations

Competing Interests

Author Contributions

Human Ethics

Data Availability

The authors declare there are no competing interests.

Guner Cicek conceived and designed the experiments, performed the experiments, analyzed the data, prepared figures and/or tables, authored or reviewed drafts of the article, and approved the final draft.

Oguzhan Ozcan conceived and designed the experiments, performed the experiments, analyzed the data, authored or reviewed drafts of the article, and approved the final draft.

Pelin Akyol conceived and designed the experiments, performed the experiments, analyzed the data, authored or reviewed drafts of the article, and approved the final draft.

Ozkan Isik conceived and designed the experiments, performed the experiments, analyzed the data, prepared figures and/or tables, authored or reviewed drafts of the article, and approved the final draft.

Dario Novak conceived and designed the experiments, performed the experiments, analyzed the data, authored or reviewed drafts of the article, and approved the final draft.

Hamza Küçük conceived and designed the experiments, performed the experiments, analyzed the data, prepared figures and/or tables, authored or reviewed drafts of the article, and approved the final draft.

The following information was supplied relating to ethical approvals (i.e., approving body and any reference numbers):

Ondokuz Mayıs University

The following information was supplied regarding data availability:

The raw data is available in the Supplemental File.

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
