# Peer review of "The effect of aerobic and high-intensity interval training on plasma pentraxin 3 and lipid parameters in overweight and obese women"

_PeerJ, doi:10.7717/peerj.18123_

## Round 0.1 · original submission · Major Revisions

Dear Authors, the manuscript covers an important and timely issue. Please address the reviewers' comments, and I will be happy to reconsider my decision.

**Language Note:** The review process has identified that the English language must be improved. PeerJ can provide language editing services - please contact us at [email protected] for pricing (be sure to provide your manuscript number and title). Alternatively, you should make your own arrangements to improve the language quality and provide details in your response letter. – PeerJ Staff

·

Basic reporting

This article employs academic and professional English.

The references are suitable for the topic, but they could benefit from some supplementation.

Certain tables require a thorough review and adjustment (I will provide a detailed description later).

The hypothesis needs to be reviewed and refined for clarity and accuracy.

Experimental design

Abstract:
In the results section, it is recommended to present the findings of the primary outcome (plasma PTX3 levels) first. Given that this is the primary outcome, it is suggested to display the changes in ng/dl, at least for this variable.

The authors mention a significant decrease in glucose, insulin, HOMA-IR, LDL-C, hs-CRP, and HDL-C; however, clarification is needed if this decrease occurred in both groups.

It is crucial to clearly describe if statistically significant differences were observed between groups (mainly between AE and HIIT) in the analyzed variables.

In the conclusions section, it is suggested to include whether differences were found between AE and HIIT in the increase of plasma PTX3 levels.

The protocol registration was not identified in the article. Was it registered in ClinicalTrials or another organization?


Introduction:
The authors mention several studies addressing the effect of exercise on PTX3 levels. It is suggested to describe what types of exercises were tested in these studies and what populations were analyzed to provide more context.

When the authors mention "there is no study yet comparing the effects of different exercise programs on PTX3 in obese individuals," do they refer to the lack of evidence on any type of exercise in people with obesity, or to the lack of comparison between AE and HIIT on PTX3 in people with obesity? This statement needs clarity and additional support.

Expanding on the arguments about why AE and HIIT were chosen over other types of exercise is recommended.

The hypothesis formulation seems more like a research question. It is suggested to review and adjust according to the principles described in https://shorturl.at/fjORY .

The objective should be adjusted to make it clear that the aim is to "compare" AE and HIIT, considering that this study was an RCT designed based on CONSORT.

The selection criteria should be revised to standardize the terms used and ensure accuracy. “sedentary (exercising ≤1–2 times/wk)” in this case, people with sedentary behavior were not included, but rather physically inactive. It is suggested to review and adjust according to https://pubmed.ncbi.nlm.nih.gov/28599680/

It is recommended to include all analyzed variables in Table 1 to observe the initial characteristics of the participants at the beginning of the study, including p-values to assess the balanced distribution between groups.

Describe in detail the randomization process, including the program used, block size, and procedures for concealing the randomization sequence and allocation to intervention groups.

The flow diagram should be adjusted to include enrollment, allocation, follow-up, and analysis. Do not confuse the terms lost to follow-up and discontinued intervention. Additionally, it should detail the final number of participants analyzed and participants excluded with their reasons.

The duration and details of the "run-in period" should be described, along with the reasons for its inclusion in the RCT. I suggest reviewing: https://shorturl.at/eBP17 Additionally, it is advisable to incorporate details of this process into the flow diagram.

It is recommended to argue why a medium effect size was used in calculating the sample size, considering the available evidence in the literature.

In the exercise training protocols, the authors solely elaborate on the HIIT group, mentioning: "...21–29 min, running." It is imperative to distinctly outline the type of exercise and the equipment used (such as treadmill, bicycle, etc.) in both groups.

Validity of the findings

Results:
It is recommended to present the results in the unit of measurement of each variable, along with their respective statistics, for better understanding and practical applicability. Hence, I propose retaining the data as presented in tables 2 and 3, while refining the description of the results (lines 234-253) in alignment with the aforementioned suggestion. This adjustment would provide readers with a more comprehensive and precise understanding.

Modify the order in which the results of the relationships between variables are described to prioritize PTX3 and its correlations.

Discussion:
It is suggested to start the discussion with the findings regarding the primary outcome (PTX3), considering its practical importance. Do the observed increases in PTX3 hold practical significance?

Keep the objective and hypothesis at the heart of the discussion, addressing important questions and comparing findings with similar studies.

Continuing with the outcomes deduced as secondary (lipid parameters and insulin resistance), and subsequently, proceeding with the remaining variables studied, I strongly recommend keeping the objective and hypothesis in mind throughout the discussion writing process, ensuring these two aspects serve as the central guiding principles. Endeavor to address questions such as: What are the most significant findings of your study? Did you reject the hypothesis? Did your findings suggest an alternative hypothesis? What are the strengths and weaknesses of your study? What other factors could have influenced your findings? How do your findings correlate with those of other relevant studies? Why do the findings of your study differ from those of other studies? What are the strengths and weaknesses of your study compared to others? I suggest reviewing: https://shorturl.at/bfgtD https://shorturl.at/joJTV. Based on the aforementioned, I recommend thoroughly reviewing the discussion section and making pertinent adjustments. This will enable readers to contextualize the findings of the research and understand its contributions to the studied problem, the practical implications of the results, and recommendations for possible future studies that should be conducted to elucidate or delve deeper into relevant aspects of the findings.

Review and adjust the discussion section to contextualize the findings of the research more clearly and systematically, as well as to highlight the practical implications and recommendations for future research.

I recommend enhancing the discussion on the outcomes of VO2max and body composition by incorporating findings from other studies comparing AE vs. HIIT in overweight and obese women. Please refer to: https://pubmed.ncbi.nlm.nih.gov/35228846/

Reviewer 2 ·

Basic reporting

.

Experimental design

.

Validity of the findings

.

Additional comments

See the attached PDF

Annotated reviews are not available for download in order to protect the identity of reviewers who chose to remain anonymous.

---

## Round 0.2 · Minor Revisions

Dear Authors:

The manuscript is really improved, but some final minor reviews must be attended,

Regards

·

Basic reporting

I have thoroughly reviewed your manuscript in detail, and I am pleased to inform you that I perceive significant improvements based on the comments I provided in the initial review.

The manuscript is written in clear and professional English throughout. The literature references provided are extensive and relevant, offering a solid background and context for your study. The structure of the article adheres to professional standards. The inclusion of figures and tables enhances the clarity of your data presentation. I appreciate the transparency in sharing the raw data. This openness is crucial for reproducibility and adds significant value to your research. Your manuscript is self-contained, presenting relevant results that are directly tied to the stated hypotheses. The discussion and conclusion sections effectively interpret these results within the broader context of the field.

Although you responded to the comment, "It is recommended to argue why a medium effect size was used in calculating the sample size, considering the available evidence in the literature," by referencing Kilic (2014) regarding the effect size, I respectfully suggest that for future studies you consider incorporating additional sources of information for sample size calculation. This should include information from previous studies, such as the minimum difference to be detected between groups, the standard deviations, the established power, and confidence levels, among other relevant parameters. Additionally, it is important to clearly determine in your sample size calculations whether the study is a superiority, non-inferiority, or equivalence RCT.

Overall, I commend you for addressing all the recommendations from the first review in a detailed and thorough manner. Your efforts have resulted in a high-quality manuscript that makes a valuable contribution to the field.

However, I recommend reviewing the entire text for possible typographical errors, spacing issues, and other minor writing errors to ensure the manuscript is polished to the highest standard.

Experimental design

No comment

Validity of the findings

No comment

Additional comments

No comment

Reviewer 2 ·

Basic reporting

.

Experimental design

.

Validity of the findings

.

Additional comments

.

Annotated reviews are not available for download in order to protect the identity of reviewers who chose to remain anonymous.

---

## Round 0.3 · Minor Revisions

Dear Co-authors:

Thank you for your patience and the changes introduced in the final manuscript. Minor modifications are required for its final acceptance in PeerJ. This decision of minor reviews is prior to being able to be published in our journal. Please take into account these minor comments from the reviewer.

Thank you

Dr. Manuel Jiménez

Reviewer 2 ·

Basic reporting

See attached PDF

Experimental design

See attached PDF

Validity of the findings

See attached PDF

Annotated reviews are not available for download in order to protect the identity of reviewers who chose to remain anonymous.

---

## Round 0.4 · accepted · Accept

Dear Authors:

I apologise for the delay; it has been a very interesting paper, and perhaps, for that reason, it has required more attention from the reviewers. I am pleased to inform you that it has been accepted for publication in its current format. I appreciate your patience, and congratulations on the work.

Sincerely

Dr. Manuel Jiménez